# Short- and Long-Term Mortality Trends in STEMI-Cardiogenic Shock over Three Decades (1989–2018): The Ruti-STEMI-Shock Registry

**DOI:** 10.3390/jcm9082398

**Published:** 2020-07-27

**Authors:** Cosme García-García, Teresa Oliveras, Nabil El Ouaddi, Ferran Rueda, Jordi Serra, Carlos Labata, Marc Ferrer, German Cediel, Santiago Montero, Maria Jose Martínez, Helena Resta, Oriol de Diego, Joan Vila, Irene R Dégano, Roberto Elosua, Josep Lupón, Antoni Bayes-Genis

**Affiliations:** 1Heart Institute, Hospital Universitari Germans Trias i Pujol, 08916 Badalona, Spain; 3aoliveras@gmail.com (T.O.); elouaddi@hotmail.com (N.E.O.); fruedasobella@hotmail.com (F.R.); jserra.germanstrias@gencat.cat (J.S.); clabata@hotmail.com (C.L.); mafema1986@hotmail.com (M.F.); gecediel@yahoo.com (G.C.); monteroaradas@gmail.com (S.M.); mj.martinez.membrive@gmail.com (M.J.M.); helena.rs92@gmail.com (H.R.); orioldediego@gmail.com (O.d.D.); jluponroses@gmail.com (J.L.); abayesgenis@gmail.com (A.B.-G.); 2CIBER Enfermedades Cardiovasculares (CIBERCV), 08080 Madrid, Spain; iroman@imim.es (I.R.D.); relosua@imim.es (R.E.); 3REGICOR Study Group, IMIM (Institut Hospital del Mar d’Investigacions Mèdiques), 08003 Barcelona, Spain; jvila@imim.es; 4Faculty of Medicine, University of Vic–Central University of Catalonia (UVic-UCC), 08500 Vic, Spain; 5Cardiovascular Epidemiology and Genetics Group, IMIM, 08003 Barcelona, Spain; 6Department of Medicine, Autonomous University of Barcelona, 08916 Barcelona, Spain

**Keywords:** ST-elevation myocardial infarction, prognosis, STEMI complications, STEMI mortality

## Abstract

Aims: Cardiogenic shock (CS) is an ominous complication of ST-elevation myocardial infarction (STEMI), despite the recent widespread use of reperfusion and invasive management. The Ruti-STEMI-Shock registry analysed the prevalence of and 30-day and 1-year mortality rates in ST-elevation myocardial infarction (STEMI) complicated by CS (STEMI-CS) over the last three decades. Methods and Results: From February 1989 to December 2018, 493 STEMI-CS patients were consecutively admitted in a well-defined geographical area of ~850,000 inhabitants. Patients were classified into six five-year periods based on their year of admission. STEMI-CS mortality trends were analysed at 30 days and 1 year across the six strata. Cox regression analyses were performed for comparisons. Mean age was 67.5 ± 11.7 years; 69.4% were men. STEMI-CS prevalence did not decline from period 1 to 6 (7.1 vs. 6.2%, *p* = 0.218). Reperfusion therapy increased from 22.5% in 1989–1993 to 85.4% in 2014–2018. Thirty-day all-cause mortality declined from period 1 to 6 (65% vs. 50.5%, *p* < 0.001), with a 9% reduction after multivariable adjustment (HR: 0.91; 95% CI: 0.84–0.99; *p* = 0.024). One-year all-cause mortality declined from period 1 to 6 (67.5% vs. 57.3%, *p* = 0.001), with an 8% reduction after multivariable adjustment (HR: 0.92; 95% CI: 0.85–0.99; *p* = 0.030). Short- and long-term mortality trends in patients aged ≥ 75 years remained ~75%. Conclusions: Short- and long-term STEMI-CS-related mortality declined over the last 30 years, to ~50% of all patients. We have failed to achieve any mortality benefit in STEMI-CS patients over 75 years of age.

## 1. Introduction

Cardiogenic shock (CS) is a low cardiac output state caused by myocardial dysfunction that leads to severe hypoperfusion resulting in life-threatening critical multiorgan failure. CS is the leading cause of hospital mortality associated with acute myocardial infarction (MI). Prevalence of CS due to acute MI varies from 5 to 15% [1,2,3,4,5,6,7,8], although some of these data come from studies performed before the generalization of reperfusion [1,2]. Despite recent advances in the prevention and management of acute MI, and the widespread use of primary percutaneous coronary intervention in patients with ST-elevation MI (STEMI), acute phase mortality of STEMI-complicated CS (STEMI-CS) remains unacceptably high [6,7,8,9,10,11]. Data on long-term trends in STEMI-CS prevalence and short- and long-term mortality are scarce. Indeed, most of the reported data reflect either limited trends, rarely beyond a decade, or only in-hospital or short-term evolution [8,9].

Accordingly, the aim of this study was to describe trends in prevalence, management, in-hospital complications, and 30-day and 1-year mortality in STEMI-CS over the last three decades (1989–2018). In addition, it also aimed to address management and prognosis across sex and age, with emphasis in cardiogenic shock patients ≥75 years during the same study period.

## 2. Methods and Materials

### 2.1. Study Population

The Ruti-STEMI-SHOCK registry is a prospective population-based registry maintained from February 1989 to December 2018. It includes all eligible STEMI patients from a well-defined geographical area of ~850,000 inhabitants in the northern metro area of Barcelona in Catalonia, Spain (Figure 1). During the 30-year period the physical healthcare structure has remained stable, with only one university hospital with an intensive cardiac care unit (ICCU) and four community hospitals that refer STEMI patients to the ICCU (Figure 1). Several organizational changes have occurred during the registry period. Until the year 2000, reperfusion therapy was mainly performed with fibrinolysis and, from 2000 to 2009, primary percutaneous coronary intervention (PCI) was performed only during working hours. A major change in June 2009 was the establishment of the “Codi IAM” STEMI network, intended as a reperfusion network that prioritizes primary PCI (pPCI) for all STEMI patients 24/7. The set-up of the Codi IAM network, including its territorial organization and available resources, has been described previously [12,13].

Definitions of myocardial infarction and the standard of care were based on current guidelines available during the study lifespan [14,15,16,17]. STEMI was defined as ST elevation of ≥1 mm in at least two contiguous leads (in V2 and V3 ≥ 2 mm was required) in any location in the index or qualifying electrocardiogram. Cardiogenic shock was defined as systolic blood pressure <90 mm Hg (after adequate fluid challenge) for 30 min or a need for vasopressor therapy to maintain systolic blood pressure >90 mm Hg, and signs of hypoperfusion (altered mental status/confusion, cold periphery, oliguria <0.5 mL/kg/h for the previous 6 h, or blood lactate >2 mmol/L) [1,18].

Patients were stratified depending on the year of admission into six five-year periods: 1989–1993 (period 1), 1994–1998 (period 2), 1999–2003 (period 3), 2004–2008 (period 4), 2009–2013 (period 5), and 2014–2018 (period 6).

All study procedures were in accordance with the ethical standards outlined in the Declaration of Helsinki. Patients provided written consent for use of their clinical data for research purposes.

### 2.2. Outcomes

The aim of the study was to analyse STEMI-CS trends in 30-day and 1-year case-fatality over the last three decades. Mortality rates were curated from patient health records and/or by direct phone contact with patients or relatives and verified by the Catalan and Spanish health system databases. Thirty-day and 1-year mortality rates for the first period (1989–1993) were incomplete and have not been incorporated in the results. Thirty-day mortality data was complete in 99% of cardiogenic shock patients and 1-year mortality data was available in 98.2% of patients.

Secondary endpoints included changes in the most relevant in-hospital STEMI-CS complications during the six studied periods: primary ventricular fibrillation and tachycardia, atrio-ventricular block, atrial fibrillation/flutter, ventricular septum or papillary muscle or free wall rupture, and right ventricle dysfunction. Definitions of these complications have remained stable during the study period. In-hospital complications were adjudicated by two independent physicians. We also analysed trends in the in-hospital management of STEMI-CS patients.

### 2.3. Statistical Analysis

Categorical variables are expressed as frequencies and percentages and continuous variables as means ± standard deviation (SD). Statistical differences between groups were compared using the Chi-squared and Student’s *t*-test, or analysis of variance including linear trend analysis. Departures from normality were evaluated using normal QQ-plots. Multivariate analysis was performed with logistic regression or proportional Cox regression models (Cox), with the following covariates: age, sex, reperfusion, anterior wall MI, and previous MI. Assumptions of the linearity of continuous variables (logistic regression and Cox) and proportionality (Cox) were tested. To assess the discrimination of the Cox models, Harrell’s C statistics was used; calibration was assessed using the Royston modification of Nagelkerke’s R2 statistic test for proportional hazards models. Trend curves were graphically fitted using polynomial regression, as they provide better fits to the nonlinear data. The period of admission was treated as a continuous measure for trend testing. Probability values <0.05 from two-sided tests were considered to indicate statistical significance. All analyses were performed using the software IBM Statistics SPSS 21 (Chicago, IL, USA).

## 3. Results

A total of 7984 consecutive patients with STEMI were included during the study period; the mean age was 61.7 years (SD 12.7), and 79.2% were men. STEMI-CS developed in 493 patients (6.2% of all STEMI). Baseline demographic and clinical characteristics between STEMI-CS and non-CS patients are shown in Table 1. Patients who developed STEMI-CS were older than nonCS patients (67.5 vs. 61.3 years, *p* < 0.001) and one-third of them were aged ≥75 years. STEMI-CS patients were more likely to be women (30.6% vs. 20.2%) and more comorbid (hypertension, diabetes, peripheral disease, and previous MI). Reperfusion therapy was less often performed in STEMI-CS (63.9% vs. 72.8%; *p* < 0.001).

Acute phase intensive cardiac care unit (ICCU) mortality was 52.6% in STEMI-CS compared to 2.2% in nonCS patients (*p* < 0.001). Similar trends were observed for 30-day (61.3% vs. 3.3%; *p* < 0.001) and one-year mortality (66.5% vs. 5.6%; *p* < 0.001).

Relative to the year of admission, STEMI-CS patients were grouped into the six five-year periods defined above: 1989–1993, n = 80; 1994–1998, n = 68; 1999–2003, n = 49; 2004–2008, n = 91; 2009–2013, n = 102, and 2014–2018, n = 103 patients. STEMI-CS prevalence trended to decline by 13% between 1989–1993 and 2014–2018 (7.1% vs. 6.2%, *p* = 0.218). Table 2 shows STEMI-CS prevalence trends relative to sex across the six temporal strata.

### 3.1. STEMI-CS Characteristics and Management during the Last 30 Years

STEMI-CS demographic characteristics and management over the six studied periods are shown in Table 2. Compared with 1989–1993, STEMI-CS patients included in 2014–2018 were older (67.6 vs. 62.4 years; *p* = 0.001) and had more history of dyslipidaemia and hypertension, but less peripheral arterial disease or previous MI (9.7% vs. 30%, *p* = 0.001).

Reperfusion therapy increased four-fold from 1989–1993 to 2014–2018 (22.5% vs. 85.4%; *p* < 0.001); with remarkably different strategies over time. From 1989–1999, reperfusion was exclusively done with i.v. thrombolytics; from 2000–2009 thrombolysis and pPCI coexisted, and from 2010–2018, since the set-up of the Codi-IAM STEMI network, all patients were treated with pPCI. The use of aspirin, clopidogrel, and other antithrombotic agents, as well as statins, increased markedly over time, as did the use of inotropes (Table 2).

Mechanical ventilation did not differ significantly between periods, and noninvasive ventilation was added in the last decade as an alternative for some patients. The use of haemodynamic support for STEMI-CS patients has evolved over time, and in the period 2014–2018 an intra-aortic balloon pump was used in 35% and ventricular assist devices in 9.7% of patients. Cardiac surgery was performed in 6.1% of patients.

Medical therapies and invasive procedures have been analysed between patients older and younger than 75 years and there were no significant differences in the use of aspirin, antithrombotic therapies, statins, reperfusion therapies, IABP implantation, ventricular assistant devices, or coronary angiography. The use of mechanical ventilation was higher in patients aged over 75 (55% vs. 36.9%, *p* < 0.001).

### 3.2. STEMI-CS in-Hospital Complications Trends

Appendix A shows in-hospital CS-STEMI complications. Ventricular malignant arrhythmias (primary ventricular fibrillation or tachycardia) developed in more than 50% of STEMI-CS patients, without significant changes over time; supraventricular tachyarrhythmias increased over time. No significant differences in mechanical complications or right ventricular dysfunction were observed in the 30-year period (Appendix A), although the limited sample size of these infrequent complications could reduce the value of these results.

### 3.3. STEMI-CS 30-Day and 1-Year Mortality Trends

STEMI-CS 30-day mortality declined from period 2 to period 6 (80.9% vs. 50.5%, *p* < 0.001) (Table 3; Figure 2), with a 9% reduction after multivariable adjustment (HR: 0.91; 95% CI: 0.84–0.99; *p* = 0.024) (Table 4). This model had a Harrell’s C statistic of 0.733 (95% CI: 0.712–0.754) and the Royston modification of Nagelkerke’s R2 statistic test showed an appropriate goodness-of-fit of the model (R2 0.26).

STEMI-CS one-year all-cause mortality declined from period 1 to period 6 (85.3% vs. 57.3%, *p* = 0.007) (Table 3; Figure 3), with an 8% reduction after multivariable adjustment (HR: 0.92; 95% CI: 0.85–0.99; *p* = 0.030) (Table 4). This model had a Harrell’s C statistic of 0.733 (95% CI: 0.715–0.751) and also exhibited an appropriate goodness-of-fit (Nagelkerke’s R2 0.26).

No differences in mortality between men and women were observed in recent years. Short- and long-term mortality in patients aged ≥75 years remained ~75%, without significant changes over time.

## 4. Discussion

The data reported here from the Ruti-STEMI-Shock registry provide long-term longitudinal trends on prevalence, management, and short- and long-term mortality in STEMI-CS in a Mediterranean cohort representing an area of 850,000 inhabitants over the last 30 years. Prevalence of STEMI-CS remains ~6.2% of all STEMIs. Reperfusion therapies and invasive procedures increased notably over time. Despite the fact that 30-day case-fatality declined by 9% in adjusted models, it remains ~50% in the current primary PCI era, without sex differences. Of note, four of five older patients with STEMI-CS, particularly those over 75 years of age, die between 30 days and one year, and this has not changed in the last three decades (Figure 4).

CS prevalence is variable in different series, ranging from 5% to 15% [1,2,3,4,5,6,7]. This reflects the lack of uniform inclusion criteria in the reported registries. Whereas the Ruti-STEMI-Shock registry included only STEMI-CS patients, CS of nonSTEMI origin were also included in French [5] and Italian [10] registries. Furthermore, patients with other aetiologies of CS in addition to acute coronary syndrome were included in a French registry [7] and in the CardShock study [6]. Considering only STEMI-CS, as in the data we report here, similar prevalence (6.6%) was observed in a recent Danish cohort study [19].

CS prevalence trends are also controversial [2,7,10,19]. In our series, CS trended to decline by 13% over three decades, similar to the results of the Worcester registry [2] and Swedish STEMI registry [3]. Indeed, in our population, CS developed in 6.2% of STEMI patients in the latest (2014–2018) period, much like the findings reported from Northern Europe in the Swedish STEMI registry [3] and Danish study [19].

Relative to management and in-hospital STEMI-CS complications, important changes happened over the past three decades, the most important being the almost universal use of reperfusion with pPCI. In the most recent period (2014–2018), reperfusion was performed in >85% of patients. This value is higher than those reported in the French registry (63%) in 2005 [5] and similar to those from the Italian study (83%) [10] in 2014. Both coronary revascularization procedures and the availability of an intensive cardiac care unit have been associated with lower mortality rates [8], although benefits in survival with the use of hemodynamic support devices (i.e., an intra-aortic balloon pump [11] or Impella support [20,21]) are as yet inconclusive. Furthermore, despite proper reperfusion, no differences in mechanical complications or ventricular arrhythmias were observed over the last 30 years. A potential explanation for the unchanging prevalence of primary VF may be that this arrhythmia occurs primarily (~75% of cases in our setting) out of hospital, before any reperfusion therapy is in place [22].

STEMI-CS crude and adjusted 30-day case-fatality declined over the last 30 years. Compared with the second period (1994–1998), in 2014–2018 we observed a 38% unadjusted relative reduction in 30-day mortality rates (from 80.9% to 50.5%). Substantially higher was the 50% 30-day mortality rate reduction shown in the Danish registry in 2017 [19]. The decline in acute phase mortality we observed was slightly more pronounced in women than in men, indicating the end of sex differences in CS-STEMI prognosis in the modern era and confirming data from other contemporary AMI registries [23]. After multivariable adjustment, 30-day case-fatality for STEMI-CS remained significant, with a 9% reduction; this improvement was mainly observed in patients younger than 75 years (a nearly 40% relative reduction in 30-day mortality). Remarkably, short-term mortality in patients aged over 75 years remained above 79% in all periods, without significant changes over time. Similar results were reproduced at one year, with an 8% adjusted decline over time and a 33% relative unadjusted decline (from 85.3% to 57.3%).

The modest achievements obtained in the past three decades do not invite complacency; outcome differences between STEMI-CS and nonCS are appalling. STEMI-CS prognosis, both short- and long-term, remains overwhelmingly poor. Different shock scores have been purposed to assess CS prognosis, the most robust being the CardShock [6] and IABP-Shock II risk scores [24] which have been validated in real-world cardiogenic shock patients [25]. These scores combine classical clinical variables (age, previous AMI or stroke, ejection fraction, and aetiology of shock) with routine biochemical data (glucose, blood lactate, and renal function). Contemporary research on the CS proteome has identified a four-protein score, the CS4P, which has shown additional value above and beyond conventional CardShock or IABP-Shock II risk scores [26]. Transcriptomics and other -omics data are under intense scrutiny to support the identification of high- vs. low-risk patients [27,28], but there is still a long way to go.

This study is not without limitations, but it has the strength of its prospective population-based registry. We acknowledge that a historical cohort study covering three decades implies changes in some definitions of cardiovascular risk factors (diabetes, hypertension, and hypercholesterolaemia) which may have affected their prevalence across periods. The definitions of myocardial infarction and cardiogenic shock have also changed slightly over time. Nevertheless, the essential diagnostic criteria for shock and STEMI remained without significant changes over the study period. On the other hand, information about bleeding, acute kidney injury, or circumstances about cardiac arrest are only available since the last period and were not included.

## 5. Conclusions

Short- and long-term STEMI-CS mortality declined over the last 30 years, but still affect ~50% of all patients. We have failed to achieve any mortality benefit in STEMI-CS in patients older than 75 years, with mortality rates close to 80%. New strategies to improve CS prognosis that apply alternative thinking and better characterization of STEMI-CS pathobiology are urgently needed.

## Figures and Tables

**Figure 1 jcm-09-02398-f001:**
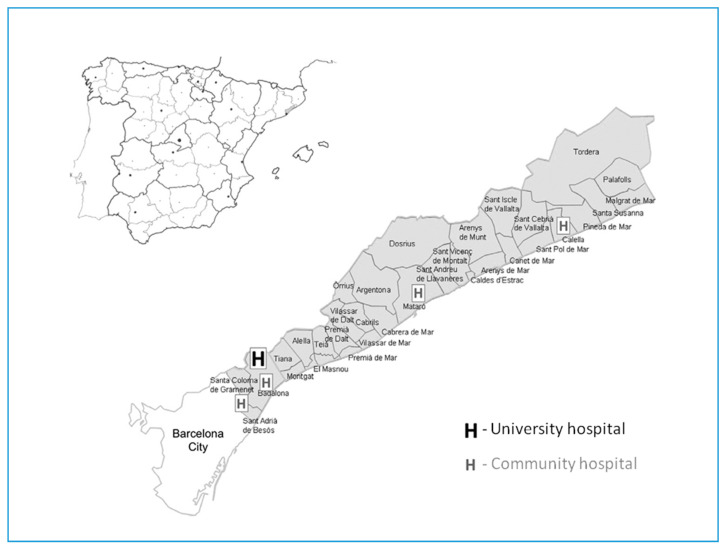
Geographical distribution of the Ruti-STEMI-Shock population-based registry in the northern Barcelona metro area.

**Figure 2 jcm-09-02398-f002:**
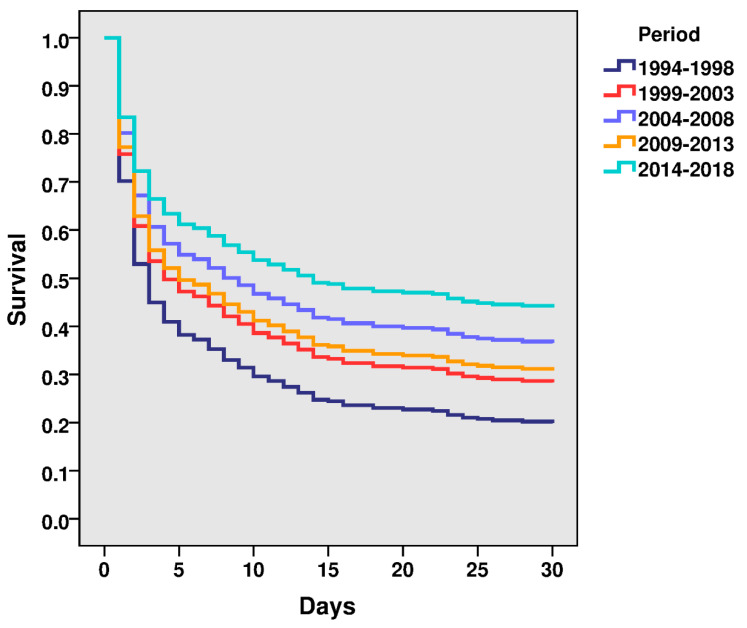
Thirty-day Cox regression multivariate analyses.

**Figure 3 jcm-09-02398-f003:**
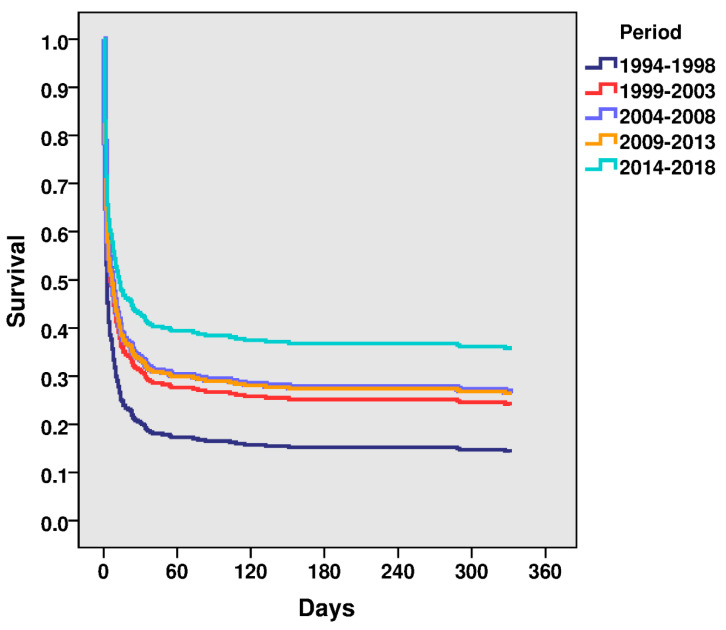
One-year Cox regression multivariate analyses.

**Figure 4 jcm-09-02398-f004:**
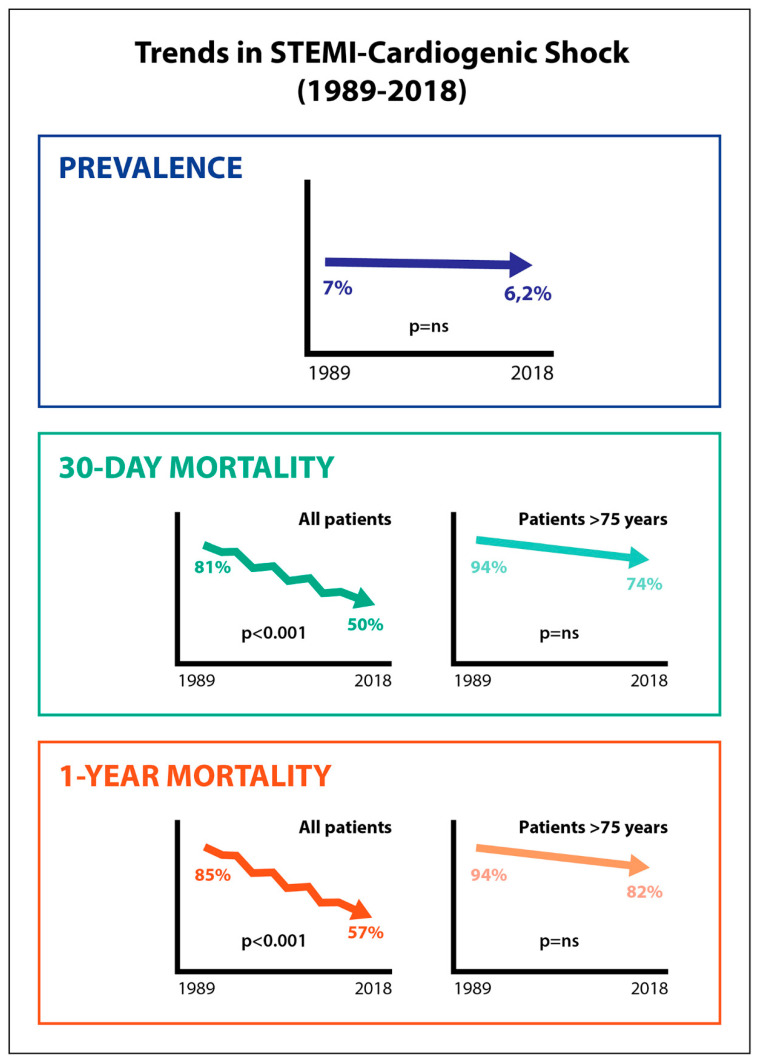
Summary illustration.

**Table 1 jcm-09-02398-t001:** Demographic characteristics, management, and prognosis in shock and non-shock STEMI patients.

Characteristics *	Whole Cohort (n = 7984)	Non-Shock Patients (n = 7491)	Shock Patients (n = 493)	*p* for Trend
Age, years: mean, (SD)	61.8 (12.6)	61.3 (12.6)	67.5 (11.7)	<0.001
Age >75 years, %	17.2	16.3	31.4	<0.001
Women, %	20.8	20.2	30.6	<0.001
Smoker, %	43.4	44.0	33.1	<0.001
Hypertension, %	51.2	50.7	58.0	0.007
Dyslipidaemia, %	51.8	52.3	43.8	<0.001
Diabetes Mellitus, %	26.3	25.8	38.1	<0.001
Peripheral Disease, %	9.7	9.4	15.2	<0.001
Previous MI, %	14.7	14.4	19.3	0.003
Anterior Wall AMI, %	44.4	43.8	51.3	0.005
Reperfusion, %	72.2	72.8	63.9	0.001
Fibrinolysis, %	31.3	31.2	31.5	0.696
Primary PCI, %	68.7	68.8	68.5	0.875
Coronary Angiography, %	56.1	56.3	52.6	0.307
Re-infarction, %	1.8	1.5	4.9	<0.001
Atrial Fibrillation, %	7.9	6.9	22.1	<0.001
Sustained VT, %	5.9	4.7	23.3	<0.001
Primary VF, %	6.8	5.6	25.2	<0.001
Complete AV Block *, %	11.9	10.3	39.1	<0.001
Right Ventric. MI *, %	11.4	9.2	50.6	<0.001
FWR, %	1.2	0.9	6.5	<0.001
PMR, %	0.4	0.2	3.2	<0.001
VSR, %	0.7	0.2	7.1	<0.001
ICCU LoS, days (SD)	3.7 (3.7)	3.5 (3.3)	5.0 (7.5)	<0.001
ICCU mortality, %	5.5	2.2	52.6	<0.001
30-day mortality, %	6.9	3.3	61.3	<0.001
1-year mortality, %	9.4	5.6	66.5	<0.001

ICCU, intensive cardiovascular care unit; AMI, acute myocardial infarction; LoS, length of stay; * only considering inferior wall AMI; VF, ventricular fibrillation; VT, ventricular tachycardia; FWR, free wall rupture; PMR, papillary muscle rupture; VSR, ventricular septal rupture.

**Table 2 jcm-09-02398-t002:** Demographic characteristics and management of STEMI-CS patients across the six studied periods.

	Period 1 1989–1993 (N = 80)	Period 2 1994–1998 (N = 68)	Period 3 1999–2003 (N = 49)	Period 4 2004–2008 (N = 91)	Period 5 2009–2013 (N = 102)	Period 6 2014–2018 (N = 103)	*p* for Trend
Age, years, -mean, (SD)	62.4 (12.0)	68.5 (11.4)	67.2 (10.7)	67.4 (11.7)	70.6 (11.1)	67.6 (11.8)	0.001
Elderly (≥75 years)	15.7	25.8	26.5	34.4	44.0	33.0	0.004
Women, %	26.3	38.3	34.7	30.8	31.4	26.0	0.560
Smoker, %	41.3	27.9	36.7	29.1	24.5	39.8	<0.001
Hypertension, %	50.0	51.5	49.0	63.7	62.7	63.1	0.145
Dyslipidaemia, %	32.5	23.5	22.4	46.2	58.8	59.2	<0.001
Diabetes mellitus, %	31.3	38.2	44.9	42.9	42.2	32.0	0.332
Peripheral disease, %	21.3	13.2	20.0	23.1	15.7	10.7	0.043
Previous AMI	30.0	29.4	24.5	16.5	13.7	9.7	0.001
Anterior Wall AMI	52.5	51.5	49.0	50.5	51.0	52.4	0.031
Prevalence of CS	7.1	7.0	5.0	6.4	5.6	6.2	0.218
Men	6.6	5.4	4.0	5.6	4.9	5.8	0.552
Women	8.9	12.8	9.6	9.3	8.2	7.6	0.147
Reperfusion, %	22.5	44.1	65.3	72.5	79.4	85.4	<0.001
Fibrinolysis, %	100	100	60.6	36.3	0	0	
Primary PCI, %	0	0	39.4	67.7	100	100	
Time onset-reperfusion *	--	--	275 (623)	252 (256)	253 (302)	185 (127)	0.027
Coronary angiography	0.0	1.5	24.5	50.5	80.4	85.4	<0.001
	**Period 1 1989–1993 (N = 80)**	**Period 2 1994–1998 (N = 68)**	**Period 3 1999–2003 (N = 49)**	**Period 4 2004–2008 (N = 91)**	**Period 5 2009–2013 (N = 102)**	**Period 6 2014–2018 (N = 103)**	***p* for Trend**
In-Hospital Management							
Aspirin, %	65.0	63.2	83.7	83.5	97.6	96.9	<0.001
Clopidogrel, %	----	----	10.2	50.5	58.8	69.9	<0.001
Ticagrelor, %	----	----	----	----	0.0	4.9	<0.001
Prasugrel, %	----	----	----	----	1.0	23.3	<0.001
IIb/IIIa inhibitors %	----	----	6.1	27.5	21.6	12.7	<0.001
Heparin, %	47.5	38.2	69.4	78.0	58.8	64.1	<0.001
Low-molecular weight heparin, %	11.3	16.2	22.4	24.2	21.6	37.9	<0.001
Statins, %	2.5	2.1	2.0	28.6	35.3	71.8	<0.001
Inotropes, %	86.3	89.7	93.9	92.3	92.5	93.2	<0.001
Invasive Mec. Ventilation, %	47.5	58.8	61.2	64.8	49.0	48.5	0.088
Non Invasive Mec. Ventilation, %	----	----	----	----	2.0	14.6	<0.001
IABP, %	----	----	12.2	35.2	38.2	35.9	<0.001
Ventricular support device (Impella CP), %	----	----	----	----	----	9.7	----
Mild Hypothermia, %	----	----	----	----	----	11.7	----
Pulmonary artery catheter, %	37.5	38.2	36.7	23.1	12.7	32.0	<0.001

SD, standard deviation; AMI, acute myocardial infarction; PCI, percutaneous coronary intervention; ACE, angiotensin-converting enzyme; ARB, angiotensin receptor blocker; IABP, intra-aortic balloon pump. * minutes, median (Interquartile rank).

**Table 3 jcm-09-02398-t003:** Acute phase, 30-day case-fatality and 1-year all-cause mortality relative to strata, sex and age.

	Period 1 1989–1993 (N = 80)	Period 2 1994–1998 (N = 68)	Period 3 1999–2003 (N = 49)	Period 4 2004–2008 (N = 91)	Period 5 2009–2013 (N = 102)	Period 6 2014–2018 (N = 103)	*p* for Trend
ICCU mortality, %	61.3	76.5	59.2	53.8	52.0	43.7	0.001
Men	59.3	71.4	62.5	50.8	50.0	43.4	0.054
Women	66.7	84.6	52.9	60.7	56.3	44.3	0.068
Young (<75 years)	62.7	69.6	50.0	44.1	37.5	33.3	<0.001
Elderly (≥75 years)	54.5	87.5	84.6	74.2	70.5	64.7	0.348
30-day case-fatality, %	NA	80.9	63.3	56.0	59.8	50.5	<0.001
Men	NA	76.2	68.8	50.8	57.1	50.0	0.006
Women	NA	88.5	52.9	67.9	65.	51.9	0.024
Young (<75 years)	NA	73.9	52.8	45.8	44.6	39.1	<0.001
Elderly (≥75 years)	NA	93.8	92.3	74.2	79.5	73.5	0.085
1-year mortality, %	NA	85.3	65.3	68.1	61.8	57.3	<0.001
Men	NA	83.3	68.8	61.9	58.6	56.6	0.004
Women	NA	88.5	58.8	82.1	68.8	59.3	0.048
Young (<75 years)	NA	80.4	55.6	59.3	46.4	44.9	<0.001
Elderly (≥75 years)	NA	93.8	92.3	83.9	81.8	82.4	0.208

ICCU, intensive cardiovascular care unit.

**Table 4 jcm-09-02398-t004:** Multivariable Cox regression analyses for ICCU mortality, 30-day case fatality and 1-year all-cause mortality.

	ICCU Mortality OR (95% CI)	*p*	30-Day Mortality HR (95% CI)	*p*	1-Year Mortality HR (95% CI)	*p*
Study Period	0.83 (0.73–0.95)	0.005	0.91 (0.84–0.99)	0.024	0.92 (0.85–0.99)	0.030
Age	1.05 (1.03–1.07)	<0.001	1.03 (1.02–1.05)	<0.001	1.03 (1.02–1.04)	<0.001
Sex	0.98 (0.63–1.51)	0.914	0.99 (0.77–1.28)	0.968	1.03 (0.81–1.32)	0.811
Reperfusion	0.59 (0.38–0.93)	0.024	0.76 (0.58–1.00)	0.052	0.78 (0.60–1.02)	0.070
Anterior Wall MI	0.95 (0.65–1.40)	0.794	0.93 (0.74–1.18)	0.570	0.93 (0.74–1.16)	0.526
Prior MI	1.09 (0.65–1.81)	0.747	0.99 (0.74–1.33)	0.961	1.00 (0.76–1.33)	0.966

ICCU, intensive cardiovascular care unit; HR, hazard ratio; MI, myocardial infarction.

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
