# Peer review of "Short- and Long-Term Mortality Trends in STEMI-Cardiogenic Shock over Three Decades (1989–2018): The Ruti-STEMI-Shock Registry"

_jcm, 2020, doi:10.3390/jcm9082398_

Round 1
Reviewer 1 Report
In this study Garcia-Garcia and colleagues utilized the Ruti -STEMI registry to analyze the temporal trends in 30d and 1 yr mortality in patients with STEMI associated cardiogenic shock (CS) between 1989 and 2018 at a single major center in Spain. Among the 493 pts with CS, the incidence was unchanged, but reperfusion therapy improved , and 30d (65%-> 51%) and 1 yr (68%->57%) mortality improved. No differences were observed in patients over 75 yrs. The strength of the data is clearly the 3 decades worth of data witch is unusual for registries. I offer the following comments.
Methods
- How were complications adjudicated and ascertained?
- What percentage of patients had complete 30d and 1 yr data? How was missingness of patients potentially moving away handled?
- The authors have used a very rigorous trial definition of CS. How was this identified in the registry?
- Did the authors consider patients with hypoperfusion without hypotension? (see Menon V et al Am J Med 2000)
Results
- The authors have stated there was not improvement in those over 75, but there is insufficient data to allow readers and reviewers to assess this finding. What was the sample size? Where patients < or > 75 treated differently?
- Do the authors have information on time to reperfusion?
- Were any patients revascularized with CABG?
- What proportion of patients had a pre- or in-hospital cardiac arrest? VT and VF are in table 1 but the breakdown of location is not clear.
- What proportion of patients had non IABP MCS support? Ie: Impella, ECMO etc
- In Table 2, how did 100% of patients undergo primary PCI, by angiography rates are < 100%?
- In the multi-variable model, what is the value in a population where such large temporal differences in reperfusion and outcomes were observed? Shouldn’t year and reperfusion be included into the model? What was the c-index and calibration of this model?
Author Response
Reviewer 1
In this study Garcia-Garcia and colleagues utilized the Ruti -STEMI registry to analyze the temporal trends in 30d and 1 yr mortality in patients with STEMI associated cardiogenic shock (CS) between 1989 and 2018 at a single major center in Spain. Among the 493 pts with CS, the incidence was unchanged, but reperfusion therapy improved and 30d (65%-> 51%) and 1 yr (68%->57%) mortality improved. No differences were observed in patients over 75 yrs. The strength of the data is clearly the 3 decade’s worth of data which is unusual for registries. I offer the following comments.
- Thank you very much for the reviewer comment, it is a good summary of our results.
- Methods
- How were complications adjudicated and ascertained?
In-hospital complications were adjudicated by two independent physicians. We have incorporated this issue in Methods section in the revised version of the manuscript.
- What percentage of patients had complete 30d and 1 yr data?
Thank you very much for the reviewer comment. Mortality was curated from patient health records and/or by direct phone contact with patients or relatives and verified by the Catalan and Spanish health system databases. Cardiogenic shock 30 days mortality data was complete in 99% of patients and 1 year vital status were available in 98.2% of patients. 30 days and 1 year data for the first period (1989-1993) were incomplete and excluded for analyses. This information has been included in the new version of the manuscript in methods section.
- How was missingness of patients potentially moving away handled?
- We appreciate the reviewer comment. The Ruti-STEMI Shock is a prospective and comprehensive registry with less than 5% of missing data in each variable. In the last 15 years this percentage was even lower.
- The authors have used a very rigorous trial definition of CS. How was this identified in the registry?
- Thank you very much for the reviewer comment.
We used the classical clinical definition of cardiogenic shock as we described in methods section. “Cardiogenic shock was defined as systolic blood pressure <90 mm Hg (after adequate fluid challenge) for 30 minutes or a need for vasopressor therapy to maintain systolic blood pressure >90 mm Hg, and signs of hypoperfusion (altered mental status/confusion, cold periphery, oliguria <0.5mL/kg/h for the previous 6 h, or blood lactate >2 mmol/L) (reference 1,18)”. This definition remained stable throughout the temporal analysis over the 30 years.
- Did the authors consider patients with hypoperfusion without hypotension? (see Menon V et al Am J Med 2000)
- We appreciate the reviewer comment but we did not consider patients without hypotension in this shock registry. Cardiogenic shock was defined following the classical definition of Shock Trial and ESC Guidelines (reference 1, 18), as we described in the method section.
- Results
- a. The authors have stated there was not improvement in those over 75, but there is insufficient data to allow readers and reviewers to assess this finding. What was the sample size?
- Thank you very much for the reviewer comment. In our series, 155 (31.4%) of cardiogenic shock patients were older than 75 years. This information had been added in each time period in Table 2.
- Where patients < or > 75 treated differently?
- We appreciate the reviewer comment. Medical therapies and invasive procedures have been analyzed between patients < or ≥ 75 years and there were no significant differences in the use of aspirin, antithrombotic therapies, statins, reperfusion therapies, IABP implantation, ventricular assistant devices, or coronary angiography. The use of mechanical ventilation was higher in patients ≥75 years (55% vs 36.9%, p<0.001). This information has been included in the results section.
- Do the authors have information on time to reperfusion?
Thank you very much for the reviewer question. Attending the reviewer‘s suggestions time from pain onset to reperfusion has been included in the new Table 2. Unfortunately, this information was not available for periods 1 and 2.
- Were any patients revascularized with CABG?
-Thank you very much for the reviewer comment. In our registry of CS patients, cardiac surgery was performed in 6.1% of patients. We have included these data in the results section of the revised manuscript.
- What proportion of patients had a pre- or in-hospital cardiac arrest? VT and VF are in table 1 but the breakdown of location is not clear.
- We appreciate the reviewer comment, but we only have comprehensive and reliable data on VT or VF location in the last period, in which 66% of VF/VT patients had out-of-hospital cardiac arrest.
- What proportion of patients had non IABP MCS support? Ie: Impella, ECMO etc
-Thank you very much for the reviewer comment. Impella CP ® became available in 2017, and it was used in 9.7% of cardiogenic shock patients. These data are shown in Table 2. In this registry there are no patients with ECMO, because it only became available recently.
- In Table 2, how did 100% of patients undergo primary PCI, by angiography rates are < 100%?
- Thank you very much for the reviewer comment. Primary PCI is a subheading of reperfusion in Table 2. In period 5 and 6 reperfusion technique was primary PCI for 100% of reperfused patients although reperfusion and coronary angiography were performed only in 80-85% of all cardiogenic shock patients.
- In the multi-variable model, what is the value in a population where such large temporal differences in reperfusion and outcomes were observed? Shouldn’t year and reperfusion be included into the model? What was the c-index and calibration of this model?
- We appreciate the reviewer comments. Reperfusion and period of inclusion (which includes year of admission) were included in the model. Moreover, attending the reviewer‘s suggestions, to assess the discrimination of the Cox models, Harrell’s C statistics was used; calibration was assessed using the Royston modification of Nagelkerke’s R2 statistic for proportional hazards models. Our multi-variable model had a Harrell's C statistic of 0.733 (95% CI: 0.712-0.754) and the Royston modification of Nagelkerke's R2 statistic test showed an appropriate goodness-of-fit of the model (R2 0.26). This information has been included in the methods and results section.
Reviewer 2 Report
The authors reported their 30-years registry experience in patients suffering from cardiogenic shock after STEMI (CS-STEMI) admitted to a tertiary-care hospital.
They classified the patients into 5-years cohorts and analyzed mortality trends at 30-days and 1-year across these six strata, adjusted with Cox multivariable analysis. Main findings were that incidence of CS-STEMI did not change over time, whereas, in patients < 75 years, mortality rate declined from period 1 (1989-1993) to period 6 (2014-2018).
The data are well presented. However, the main and substantial limitation of this work is its mere descriptive nature. The epidemiological characteristics of CS-STEMI are well defined by several retrospective and randomized studies performed in the last 10 years, and the present study does not add any new information in this field.
Indeed, the decrease of CS-STEMI mortality after the widespread introduction of primary PCI is almost universally reported, as well as the limited impact of therapeutic strategies beyond revascularization observed in the last two decades.
Specifically:
- the authors should better specify the aims of the study, as the only objective of epidemiology and outcomes description does not meet the current knowledge gap about CS-STEMI.
- Many outcomes measure are lacking: bleeding, AKI, end-organ damage, circumstances of cardiac arrest (OHCA etc.), type of mechanical circulatory support employed, native heart outcomes etc etc.
- AMI-related mechanical complications did not decline over time: this point must be discusses.
- No data about treatment of mechanical complications; no data about patients eventually referred for cardiac surgery.
- Lastly, the authors did not perform any analysis of the predictors of negative outcomes across the study period. The identification of risk factors for mortality over 30-years could be interesting, as it could suggest possible undressed therapeutic targets.
Author Response
Reviewer 2
- The authors reported their 30-years registry experience in patients suffering from cardiogenic shock after STEMI (CS-STEMI) admitted to a tertiary-care hospital.
They classified the patients into 5-years cohorts and analyzed mortality trends at 30-days and 1-year across these six strata, adjusted with Cox multivariable analysis. Main findings were that incidence of CS-STEMI did not change over time, whereas, in patients < 75 years, mortality rate declined from period 1 (1989-1993) to period 6 (2014-2018).
- Thank you very much for your comments.
- The data are well presented. However, the main and substantial limitation of this work is its mere descriptive nature. The epidemiological characteristics of CS-STEMI are well defined by several retrospective and randomized studies performed in the last 10 years, and the present study does not add any new information in this field. Indeed, the decrease of CS-STEMI mortality after the widespread introduction of primary PCI is almost universally reported, as well as the limited impact of therapeutic strategies beyond revascularization observed in the last two decades.
- We appreciate the reviewer comment. Although this is a descriptive observational study, this is a comprehensive registry among 30 years which confers it an important epidemiological interest. Furthermore, we analyzed differences in short- and long-term cardiogenic shock prognosis between elderly (over 75 years) and younger patients, not previously delineated with detail in other cohorts.
- Specifically:
- The authors should better specify the aims of the study, as the only objective of epidemiology and outcomes description does not meet the current knowledge gap about CS-STEMI.
- Thank you very much for the reviewer comments. Attending the reviewer‘s suggestions the aims of the study have been slightly modified.
- Many outcomes measure are lacking: bleeding, AKI, end-organ damage, circumstances of cardiac arrest (OHCA etc.), type of mechanical circulatory support employed, native heart outcomes etc etc.
- We appreciate the reviewer comments. Unfortunately, information about bleeding, AKI or circumstances of cardiac arrest are only available in the last period and we did not include it in the manuscript.
Mechanical circulatory support was available since 2017 with Impella CP ® . We used this MCS support in 9.7% of cardiogenic shock patients in this period. This data is shown in table 2. These limitations have been included in the limitations section.
- AMI-related mechanical complications did not decline over time: this point must be discusses.
- Thank you very much for your comments. Attending the reviewer‘s suggestions, discussion has been modified to clarify this point.
- No data about treatment of mechanical complications; no data about patients eventually referred for cardiac surgery.
We appreciate the reviewer comments. In our series, surgical repair was performed in 28.9% of patients with mechanical complications, achieving 56.5% in papillary muscle rupture and 15.6% in free wall rupture patients.
If we consider all cardiogenic shock patients (with and without mechanical complications), cardiac surgery was performed in 6.1% of them. This information has been incorporated in the revised version of the manuscript.
- Lastly, the authors did not perform any analysis of the predictors of negative outcomes across the study period. The identification of risk factors for mortality over 30-years could be interesting, as it could suggest possible undressed therapeutic targets.
- Thank you very much for the reviewer comments. A multivariable cox regression analyses were performed and detailed in Table 4. In these analyses, age and the lack of reperfusion were the main risk factors for 30-days case fatality. These results were confirmed in the 1-year mortality analysis.
Round 2
Reviewer 1 Report
The authors have adequately addressed my questions and concerns.
Reviewer 2 Report
- The authors reported their 30-years registry experience in patients suffering from cardiogenic shock after STEMI (CS-STEMI) admitted to a tertiary-care hospital.
They classified the patients into 5-years cohorts and analyzed mortality trends at 30-days and 1-year across these six strata, adjusted with Cox multivariable analysis. Main findings were that incidence of CS-STEMI did not change over time, whereas, in patients < 75 years, mortality rate declined from period 1 (1989-1993) to period 6 (2014-2018).
- Thank you very much for your comments.
- The data are well presented. However, the main and substantial limitation of this work is its mere descriptive nature. The epidemiological characteristics of CS-STEMI are well defined by several retrospective and randomized studies performed in the last 10 years, and the present study does not add any new information in this field. Indeed, the decrease of CS-STEMI mortality after the widespread introduction of primary PCI is almost universally reported, as well as the limited impact of therapeutic strategies beyond revascularization observed in the last two decades.
- We appreciate the reviewer comment. Although this is a descriptive observational study, this is a comprehensive registry among 30 years which confers it an important epidemiological interest. Furthermore, we analyzed differences in short- and long-term cardiogenic shock prognosis between elderly (over 75 years) and younger patients, not previously delineated with detail in other cohorts.
- Specifically:
- The authors should better specify the aims of the study, as the only objective of epidemiology and outcomes description does not meet the current knowledge gap about CS-STEMI.
- Thank you very much for the reviewer comments. Attending the reviewer‘s suggestions the aims of the study have been slightly modified.
The aims are now clearer
- Many outcomes measure are lacking: bleeding, AKI, end-organ damage, circumstances of cardiac arrest (OHCA etc.), type of mechanical circulatory support employed, native heart outcomes etc etc.
- We appreciate the reviewer comments. Unfortunately, information about bleeding, AKI or circumstances of cardiac arrest are only available in the last period and we did not include it in the manuscript.
Mechanical circulatory support was available since 2017 with Impella CP ® . We used this MCS support in 9.7% of cardiogenic shock patients in this period. This data is shown in table 2. These limitations have been included in the limitations section.
The lack of these data, unfortunately, reduces in my opinion the informative potential of this work.
- AMI-related mechanical complications did not decline over time: this point must be discusses.
- Thank you very much for your comments. Attending the reviewer‘s suggestions, discussion has been modified to clarify this point.
- No data about treatment of mechanical complications; no data about patients eventually referred for cardiac surgery.
We appreciate the reviewer comments. In our series, surgical repair was performed in 28.9% of patients with mechanical complications, achieving 56.5% in papillary muscle rupture and 15.6% in free wall rupture patients.
If we consider all cardiogenic shock patients (with and without mechanical complications), cardiac surgery was performed in 6.1% of them. This information has been incorporated in the revised version of the manuscript.
I apologise for the lack of clarity. Mechanical complications of AMI in your cohort affected a significant number of patients over time. Given the high mortality rate carried by these complications, it could be worth giving more data. The data about surgical repair of mechanical complications (reported in the response above) are lacking in the revised version, and are not adequately discussed (the lack of decline over time is an important finding that cannot be solely interpreted as related to the low number of events).
- Lastly, the authors did not perform any analysis of the predictors of negative outcomes across the study period. The identification of risk factors for mortality over 30-years could be interesting, as it could suggest possible undressed therapeutic targets.
- Thank you very much for the reviewer comments. A multivariable cox regression analyses were performed and detailed in Table 4. In these analyses, age and the lack of reperfusion were the main risk factors for 30-days case fatality. These results were confirmed in the 1-year mortality analysis.
Many potential prognostic factors have not been incorporated into the model, and the informations that can be drawn are limited.
In conclusions: I am convinced that the authors performed a commendable job in reporting CS-STEMI outcomes across a large time range. However, the only strong finding of this article is that revascularisation changed the prognosis of STEMI pts: this is by now universally recognised. The worse prognosis of elderly patients with STEMI is equally well known. Therefore, in the absence of more data and subsequent analyses and interpretations that could increase the informative value of this work (AKI-Bleeding-hypoperfusion markers-echo data-hemodynamics etc), I believe that this article did not overcome the limitations highlighted in the first revisions.